# Nurses’ Perceptions of Patient Fibromyalgia Illness Experiences after Performing Group-Based Problem-Solving Therapy: A Qualitative Research Study

**DOI:** 10.3390/healthcare11111531

**Published:** 2023-05-24

**Authors:** Pilar Montesó-Curto, Maria Luisa Panisello-Chavarria, Lidia Sarrió-Colás, Loren Toussaint

**Affiliations:** 1Primary Care in Institut Català de la Salut (ICS), 43500 Tortosa, Spain; 2Faculty of Medicine, Rovira i Virgili University, 43201 Reus, Spain; 3Department of Nursing, Faculty of Nursing, Rovira i Virgili University, 43500 Tortosa, Spain; 4Pere Mata Foundation Terres de l’Ebre, 43549 Amposta, Spain; 5Department of Psychology, Luther College, Decorah, IA 52101, USA

**Keywords:** fibromyalgia, illness perceptions, qualitative study, focus groups, nurses

## Abstract

Fibromyalgia patients experience difficulties in their daily lives that are difficult to identify and recognize due to the stigma associated with the disease. Nurses can help identify them to establish biopsychosocial coping and treatment. The main aim of this study was to explore Spanish nurses’ perceptions of the illness experiences of their fibromyalgia patients. Qualitative content analysis from the etic perspective was used. Eight nurses met in focus groups to report their perceptions of the illness experiences of FM patients after led group-based problem-solving therapy in fibromyalgia patients. Four themes emerged: (1) the presence of a “specific trigger” (stressful event) for FM symptoms; (2) fulfilling expected gender roles; (3) a lack of support from the family; (4) abuse. Nurses recognize the mind–body connection after the impact of stress on patients’ bodies. The expected gender roles interfere with patients’ recovery because they feel frustration and guilt about not being able to fulfil them. Managing emotions and improving communication in fibromyalgia is recommended. Clinicians might also consider issues such as abuse and the absence of social–family support for the comprehensive evaluation and effective management of fibromyalgia.

## 1. Introduction

Fibromyalgia (FM) is a complex syndrome with a prevalence that ranges from 2% to 6% of the world’s population [1] and is associated with high healthcare and societal costs [2]. Another important aspect is the strong predominance of female patients, who represent 80–96% of all FM patients [1]. FM patients present a wide range of symptoms: generalized widespread chronic pain, cognitive problems (fibrofog), stiffness, unrefreshing sleep, fatigue, and distress [3]. Although pain is the main symptom of FM, other symptoms are clinically significant for patients and are sometimes more disabling than pain. The ACTION-APS Pain Taxonomy (AAPT) diagnostic criteria include two other symptoms, fatigue and sleep problems, which are most commonly reported by FM patients [4]. In addition, nocioplastic pain, which arises from altered nocioperception, without evidence of tissue or somatosensory damage causing the pain, is recognized in FM and provides a new opportunity for the early diagnosis of FM beyond symptom-based scoring criteria [5].

Currently, there is no cure for FM. The best way to treat FM symptoms is to adopt a multimodal approach, with 36 pharmacological and non-pharmacological strategies tailored to each patient [6]. In her recent study, Menzies [7] describes the 37 latest treatment regimens, pharmacological approaches, and non-pharmacological approaches, including exercise; cognitive behavioral therapy (CBT); education, including alternative medicine; and nursing approaches designed to improve patient self-management. As the complete remission of symptoms is rare, and the adverse effects of medications can complicate the symptoms, it is essential for nurses to be well informed about this complex syndrome. Currently, recommended pharmacological strategies for the treatment of FMS symptoms include, among others, the use of selected anticonvulsants and pregabalin, selective serotonin reuptake inhibitors (SSRIs), selective serotonin and norepinephrine reuptake inhibitors (SNRIs), tricyclic antidepressants, and muscle relaxants. Drugs commonly used to treat peripheral pain, including non-steroidal anti-inflammatory drugs (NSAIDs) and potent opioids such as corticosteroids, are ineffective in the treatment of FMS symptoms. Non-pharmacological therapies include walking; progressive muscle relaxation; activity pacing; exercise; CBT including cognitive restructuring, guided imagery, and communication; assertiveness skills; autogenic relaxation training; cognitive reconceptualization; education on FM and the benefits of exercise; barriers to behavior change; acupuncture; hypnosis; mindfulness-based interventions such as meditation; and meditative movement therapies such as qigong, tai chi, and yoga [7].

There is a high level of uncertainty regarding FM diagnosis, pathophysiology, and treatment efficacy [8]. This uncertainty affects work, family, and social relationships and can lead to the stigmatization of FM patients [9]. Although the symptoms of FM frequently persist, many patients are able to identify coping mechanisms over time that can moderate symptoms [5]. Symptoms can be improved by changing perceptions of illness (i.e., modifying perceptions of the disease by adapting attitudes, emotions, and behaviors formulated by cognitive behavioral therapies) [10,11]. The best results were obtained after a combination of training protocols and aerobic exercises. Additionally, the intensity should be gradual and progressive [12,13]. Resistance training is a type of physical exercise that has beneficial effects for patients with FM, decreasing pain and fatigue and increasing strength and functional capacity [14]. Patients appear to perceive themselves as victims of an undiscovered disease and continue to seek medical help in anticipation that symptoms will be validated by a diagnosis discovered via medical tests [15]. A meta-ethnographic review of 77 papers showed that chronic-pain patients were struggling to affirm self, reconstruct self in time, construct an explanation for suffering, and prove legitimacy. However, despite this struggle, there was a positive sense of moving forward (i.e., listening to one’s body and not fighting against it) [16,17].

In a recent study, nurses acquired knowledge of the problems experienced by FM patients by conducting group-based problem-solving therapy (GPST), which is a cognitive behavioral therapy that uses a variety of techniques to identify and solve problems, increase assertiveness and self-esteem, and eliminate negative thoughts [18]. It is a non-specialized therapy that is relatively short and straightforward and can be offered to patients and families. Nurses are an integral part of the multidisciplinary management programs usually recommended for this syndrome, and there is a large body of evidence supporting their key role [19,20]. GPST could help FM patients to express feelings of anger since it is well known that the healthy expression of anger can alleviate the effects of FM [18]. They used this therapy to try to modify fear, anxiety, and the avoidance of activity, which are factors that often sustain pain in FM patients [21]. Patients with FM are characterized by higher levels of anger rumination (i.e., internalized) than healthy and clinical controls, and there is evidence to show that the level of pain is reduced when anger is expressed [22]. Thus, van Middendorp [23] found the level of pain to be lowest among FMS patients who tended to express their anger in an anger-arousing situation.

How FM-related experiences are perceived by clinicians has an impact on how well these problems and symptoms are managed. In other words, the accurate (non-biased) perception of patients’ experiences seems to be a prerequisite for good management [24,25]. According to Peterson [26], nurse practitioners are in a good position to help identify FM patients or those with an FM diagnosis that do not experience any relief in symptoms. The main aim of this study was to explore Spanish nurses’ perceptions of the illness experiences of FM patients after group-based problem-solving therapy (GPST) led by nurses.

## 2. Materials and Methods

### 2.1. Study Design and Sample

This paper provides an analysis of nurses’ perceptions of the illness experiences of FM patients identified in two 90 min focus groups with 8 nurses who had previously led six GPST groups of FM patients (43 female and 1 male) [27]. Seven were nursing professionals, and one was a nursing student. They were all women with a mean age of 50 years, most of whom had been working as nurse practitioners for more than thirty years (see Table 1). In this study, we have incorporated the etic perspective. The emic/etic distinction is one of the principal concepts guiding qualitative research. An emic perspective is the insider’s view of reality, while the etic perspective is the external social scientific perspective on reality. The etic view involves stepping back from the insider’s view. An external view without an emic or external foundation is unusual and is uncharacteristic of qualitative work [28].

As shown in Table 2, the socio-demographic and clinical profile of the typical FM patient was a woman, approximately 61 years of age, retired at the time of enrolment, diagnosed with FM for approximately 10 years, moderately to highly satisfied with the health system, and more satisfied with non-pharmacological treatment than with medication.

The questions of the focus group were as follows: 

(1) What are the main problems you identify in people who have suffered from fibromyalgia? They can be both problems that triggered the disease or arising from any sphere of life and/or disease.

(2) What aspects of life could be solved to improve these problems?

The study was approved by the Ethics Committee at Joan XXIII Hospital in Tarragona (Catalonia, Spain), certificate: 40P/2012.

### 2.2. Data Collection

We explored eight nurses’ perspectives and opinions on the illness experiences of FM patients via focus groups. The focus groups were conducted at a university in NE Spain since the research team consisted of members of the University’s Faculty of Nursing. In a focus group, individual views are clarified by interactions with other people, which do not occur in an interview. Therefore, it seemed preferable to use focus groups since one person’s opinion could reinforce and/or clarify that of the other(s) [29]. Both focus groups were audio recorded and transcribed by a research assistant.

### 2.3. Data Analysis and Rigor

Content analysis was used to analyze the information. Conventional content analysis is generally used for a study design with the aim to describe a phenomenon; in this case, the aim is to explore nurses’ perceptions of the illness experiences of their fibromyalgia (FM) patients [30]. Open coding was used initially (that is to say, data were coded only if relevant to the aims of the study). Three members of the research team repeatedly read the transcripts to obtain an initial awareness. The content analysis of the transcripts was then independently conducted by the first, second, and third team members. They generated codes and grouped them into categories. These were further agreed upon by all researchers in the team. These categories were then grouped into themes with approval of the whole group. Thus, the researchers avoided using previous categories. Rather, categories were encouraged to emerge from the data by researchers immersing themselves in the data so as to allow new ideas to emerge [31]; this is also described as inductive category development [30]. Themes were then generated from the perspectives of the focus group members, and it was the more abstract concepts that reflected the interpretation of patterns across the data [32]. So, from initial codes, categories were created; from categories, broader themes were established to précis the data, while retaining the richness, depth, and context of the original data [33]. The analysis was regarded as complete when no new themes were identified and when an agreement was reached among the authors [29].

The number of participants was considered to be suitable for a qualitative study since it involved a large volume of in-depth data [34]. It was considered that saturation existed when the data obtained, instead of providing new explanatory elements, only increased the volume of them. This is similar to the concept known as “theoretical saturation” [35].

## 3. Results

Most of the results are related to the importance of nurses’ awareness of the negative and painful experiences that patients with FM usually suffer and how these affect their health. A summary list of the themes identified during the analysis of the nurses’ focus-group data, with attendant categories, can be found in Table 3.

### 3.1. Theme 1: How Stress and Negative Emotions Can Affect the Body

#### 3.1.1. It Was Common for Nurses to Find the Presence of a “Specific Trigger” (Stressful Event) for FM Symptoms in Their Patients

Nurses described how some patients mentioned a complicated birth, while others mentioned a serious accident (Nurses 03 and 04), concern about their illness or that of a family member—such as cancer affecting a brother or sister—(Nurses 01 and 02), or episodes of abuse or maltreatment during childhood (Nurses 03 and 04). Deaths in the family (unresolved grief) were also mentioned (Nurses 01 and 02). Nurse 03 said: “One woman reported an accident in which her mother, daughter, brother, and niece all died, and that she has been unable to get dressed ever since. The day after the accident in which several relatives died, she could not even get dressed”. Nurses 01, 02, and 04 said: “One woman lost her fiancé just months before she was due to get married. Another lost her sister, and another her mother”.

Nurse 01 stated, “These were identified as events moments of great suffering before the onset of FM”. Nurse 01 also stated: “One woman reported a relationship problem with her children—a daughter of 20 and a son of 45. Her son was separated, heavily in debt, and had a mortgage that she was paying down, so the mother was paying her son’s bills and attempting to solve all his problems”. Nurse 04 expressed: “They had been through difficult times and, although they had changed their family status or their partners, things had not improved. They’ve gone from bad to worse”.

Nurse 05 said that there was a man in her group, the only man in the research project, who said that FM had emerged after a long period of problems at work. When he was finally fired, he felt relieved because his job was making him feel extremely anxious and he had to take a lot of medication.

During their lives, many of the patients reported having suffered serious psychological abuse and humiliation. Nurse 04 described a session in which a woman spoke about her husband, who was attracted to another women, but after revealing this information she clammed up. Nurse 04 remarked: “The psychological abuse went on throughout her life and she felt humiliated, but she acted as though nothing were going on. She thought the problem was deeply entrenched and had no solution. Her husband had been unfaithful and though she said she had forgiven him, in fact she had not”.

Nurse 01 said that one woman was so afraid of her son-in-law that she felt impotent and helpless. She did not want to say anything in case the people in her village find out, but she is aware that practically everybody knows anyway. The son-in-law, who has a tumor, manipulates the situation, and even goes so far as to hit the woman’s son. Nurse 05 stated: “In general, patients’ fear makes their clam up. They view therapy as a way to opening a door into a situation for which they believe a solution will never be found. Nurse 06 expressed: “They cover things up—first all their husbands’ flaws and then those of their children. Although, in their relationships, some women behave in such a way that they are susceptible to being mistreated, it is difficult for them to change because that is how it has always been”. All nurses concluded: “All the participants have had very complicated experiences with parents, husbands, abuse, accident… In short, a very complicated life. They all have a disturbing element behind them” (Nurse 01, 02, 03, 04, 05, 06, and 07).

#### 3.1.2. Some FM Patients Did Not Report a Specific Trigger but Did Report That They Have Had to Endure Harsh Conditions throughout Their Lives

Nurses 03, 04, and 07 said: “Some have had to work extremely hard after a separation, while others have had to care for their parents, parents-in-law or children, fulfilling their roles as caregivers and having very little time to relax “. All nurses noticed that “The patients have endured very difficult lives from both physical and emotional points of view. They have always worked hard since they were little doing physically demanding jobs” (Nurses 01, 02, 03, 04, 05, 06, 07, and 08).

#### 3.1.3. Patients Consciously/Unconsciously Highlight the Importance of Stress in FM

Nurses 03 and 04 declared: “On the days they have more problems; they say they also feel more pain”. Nurse 01 said: “They continued to relate all life problems to their mental state”. Nurses 03 and 04 stated: “General practitioners and some of their friends do not recognize their problem. What annoys them most is when they are told to, as if their problem was all in their minds and they could easily sort it out. They labelled it as a ‘psychosomatic problem. All the adversities they have endured have made them become sick”. Nurse 06 said: “For example, my mother, who always argued with my father, was always sick. For me, illness is a way to survive, to move forward”. Nurse 01 mentioned: “She have been told by a very good doctor that when she solves the difficulties of people with fibromyalgia, it can be healed”.

#### 3.1.4. Patients Were Afraid to Express What They Feel

Nurses 04 and 06 said that, when their patients did put their problems into words, they realized that they could tackle them. They said: “They have to say what they want and learn to say it. I think it’s a kind of self-analysis. They realize that the role in mistreatment is one that they have learned”. Nurse 06 explained: “I had a woman explain to me her husband’s lifelong infidelity and what she had experienced. She had never mentioned it to anybody. She did not seek therapeutic help or any help at all, but her body got sick”. Nurse 04 affirmed this woman’s feelings: “She did not come back to the other sessions. I think she felt ashamed”.

#### 3.1.5. Gradually the Patients Begin to Discuss the Role Played by the Mind and How Their Emotional State Influences Their Pain

The relationship between mind and body appears to be a constant component in the discourse of these women. Nurses had to be careful, however, because in the first GPST session any mention of “mental aspects” was largely rejected as patients felt they were being told they had a psychosomatic disorder.

Nurses 03 and 04 expressed: “At first the patients only saw the physical manifestations of their illness—how it affected their bodies. Later they began to understand how their condition was affected by their emotions and started to acquire greater expertise. With this greater expertise, they began to accept their condition and to associate it more with mental phenomena or emotions. As a result, the difficulties caused by their condition were better resolved”. Nurse 03 mentioned: “When they have an important problem but see that it has a solution, what was bothering them is no longer a problem and does not affect them physically or psychologically”.

### 3.2. Fulfilling Expected Gender Roles

#### 3.2.1. All Patients Put a Very High Limit on Their Daily Activities Because They Have to Assume Their Assigned Gender Role

All nurses reported that their patients described having to be ‘able to do everything’, relating their illness to their very difficult lives. They have never given themselves any free time and have always looked after their parents and partners (Nurses 01, 02, 03, 04, 05, 06, 07, and 08). Nurse 01 affirmed these sentiments: “They say they have never been able to rest. They find it difficult to ask for help and blame themselves. They also say that nobody does the chores as well as they do”. Nurse 02 stated: “They are very demanding of themselves”. Nurses 03 and 04 thought that these women (their patients) are exposed to a very high level of domestic demands because their traditional upbringing obliges them to comply with the gender mandate.

All nurses observed that was very difficult to change their mentality about having to do the household chores themselves. They feel guilty about not fulfilling what they consider to be their role, perceive their illness in terms of things they find difficult to accomplish and complain of feeling lazy and dirty when they are unable to take care of their home or of themselves. Some of them have started to rest more, know their limitations, and go as far as they can because if they do not do it this way the next day, they have more fatigue and pain (Nurse 01, 02, 03, 04, 05, 06, 07, and 08).

#### 3.2.2. The Patients Had Difficulty in Facing Problems

Nurses 02, 03, and 04 mentioned: “They never know how to say no. They find it difficult to ask for help and feel guilty when they need to do so, saying they have always done the chores and do not want to ask anybody for help because nobody does them as well as they do”. Nurse 01 told them that housework should be a shared responsibility; sharing those tasks doesn’t mean they are doing everything for you because everybody is living in the same house, and everyone is making it untidy. The positive aspect of the GPST is that the patients felt they were being supported and listened to. It also appeared to help them become aware of the opinions and feelings they had made subconsciously.

Nurse 06 declared: “Patients reported problems with sisters, partners and parents, and a pattern of mistreatment became apparent. However, when they could have set limits on the situation, they were incapable of doing so. One woman, who was being manipulated by her sister, shrugged her shoulders, and slumped in her chair when speaking about the situation, but did not appear to realize she was doing it”. Nurse 06 also shared: “How the women faced up to their problems depended on how long they had had the illness. Those who had had the illness for several years managed it more easily and accepted it better. We could regard them as ‘experts by experience’. They’ve got this illness and They’ve accepted it now; They’re going to try to live well now”. All nurses noticed: “They are afraid to express what they feel. They have suffered severe psychological abuse throughout their lives and have been annulled as people” (Nurse 01, 02, 03, 04, 05, 06, 07, and 08).

While conducting the GPST, nurses 02, 03, and 06 reported that they found considerable differences in the women’s self-esteem. Whether they accept the illness depends on their level of support and expertise. With time, they have begun to look for approaches that can help them overcome the discomfort caused by their illness. Nurses 01, 02, and 04 explained the case of one of the women, who had almost been confined to a wheelchair but was now feeling much better after all the treatments and solutions she had incorporated into her lifestyle. As causal factors of her FM, she identified the death of her mother when she was 20 and the death of her boyfriend when she was 23. These stressful events led her to change both her lifestyle and her attitude, and she became more relaxed. Today, she walks every day for an hour and is less self-demanding. This patient provided positive reinforcement for all the women in her group sessions and became an example of positive coping (as reported by Nurses 01, 02, and 04). Nurse 06 said: “With patients like her, one of two things can occur: either the initial problem or unknown trigger is solved, or one problem leads to another, and then another and so on “. Nurses 01 and 03 stated: “Fibromyalgia as well as in other diseases if we would work on self-esteem, it would improve considerably. You are valuable just as you are. There should be an important educational change since childhood. Holistic concepts of health should be introduced, as well as how to deal with the complexity of life and develop coping strategies so that we do not get sick”.

### 3.3. Theme 3: Lack of Support from the Family

There were differences among the patients with regard to family support, but, in general, the families showed little support.

#### 3.3.1. Some of Them Showed Some Support

Nurse 01 mentioned: “Some patients reported strong support, especially those who had recently started new relationships”.

#### 3.3.2. In General, They Showed Little Support

However, although some patients reported enjoying the support of their family, this does not necessarily mean that they actually received support. Nurses 04, 06, and 07 summarized these conflicting reports: “I think that in my group two or three patients had such support, but I cannot see that any of the others did”. Some patients said that they do not receive help and also feel overwhelmed and that the workload is enormous. All nurses stated: “Women are expected to multitask from childhood, including doing the housework and caring for a family (Nurses 01, 02, 03, 04, 05, 06, 07, and 08). Nurse 04 expressed: “She had [to make] food every day for 7 people and She is 74 years old”. Nurse 01, 05, and 06 declared: “Some have mentioned that, if they ask for their partner’s help, that they no longer take things so badly and see the glass as more full than empty”.

### 3.4. Abuse

Abuse is a phenomenon that appears in all groups.

#### 3.4.1. The Patients Experienced Physical, Psychological, and Economical Abuse

Nurses were often told about physical and/or psychological abuse. In most cases, nurses related the onset of FM to trauma during childhood, adolescence, or later (intimate partner violence). Nurse 04 described that a woman had suffered physical and psychological abuse but that she could not leave home for economic reasons. Nurse 04 verbalized: “She had no possessions after being married for over 40 years”. Nurse 03 reported that another woman confessed that her husband mistreated her from the first day of marriage. She reported: “The abuse was psychological. She just cried. She was unfaithful. She came to think that everything was normal and that she was to blame”.

Nurse 06 testified that the abuse was also economic. She explained: “Her son is desperately asking for money for his huge debts, and she has had to sell many of her belongings to help him”.

#### 3.4.2. Some Patients Have Found Ways to Manage and Protect Themselves from Abuse

Nurse 04 declared: “Some have learned to improve communication, for example writing a letter with what they think, others have learned to protect themselves and to protect themselves and to look out for themselves and not only for others”.

## 4. Discussion

Nurses recognize ‘stressful life elements’ as underpinning or triggering FM. The description of onset or trigger events often evoked emotionally painful, distressing, or shocking experiences. We have provided examples of how one’s emotions can affect the body and even paralyze it (Theme 1). Cedraschi et al. [36] identify that there are probably numerous distressing events that can have this effect, and many narratives talk of a cascade of disruptive events, suggesting an increasing loss of control. Ablin and Buskila [37] discuss the impact of stressful events, such as early life trauma. Sallinen, Kukkurainen, and Peltokallio [38] reported that the onset of FM was perceived as the result of suffering from violence but that this violence is often still hidden behind a wall of silence. The nurses in our study also found that their patients perceived that stressors increased pain. Malin and Littlejohn [39] presented similar results in their review. FM can be triggered by the experience of one or more traumatic physical and/or emotional events, which generate stress and distress in individuals. Susceptibility to stress differs from one person to another. Many of these participants’ attributes related to the onset of fibromyalgia have support in the literature; as reflected in them, causality emerges as complex and heterogeneous [40]. Patients have identified flares that worsen their symptoms during that time. The mean number of fibromyalgia flares was two per patient. The most frequent triggers identified were continuous stress (56%), intense stress (39%), physical overexertion (37%) and climatic changes (36%) [41]. On the other hand, FM patients even have higher rates for post-traumatic stress spectrum symptoms, the worsening of symptoms, and poorer quality of life [42].

Psychological stressors have a causal effect on the development of chronic pain. Clinical evidence has shown that there was an excess of LPC16:0 in FM cases, and LPC16:0 expression correlated with pain symptoms in patients with elevated oxidative stress and disease severity. Excessive oxidative stress, caused by the exposure to stress-induced lipids and lipid oxidation and the subsequent upregulation of LPC16:0, activates nociceptive signaling to cause chronic hypersensitivity through activation of acid sensing ion channel 3. Darapladib and antioxidants alleviated stress-induced chronic hyperalgesia by inhibiting LPC16:0 synthesis [43]. In this way, the application of hyperbaric oxygen therapy (HBOT), which is the application of hyperbaric pressure in conjunction with increased oxygen content, has been shown in several clinical studies to have the capacity to induce neuroplasticity that leads to the repair of persistent impaired brain functions even years after an acute injury [44].

The women told their nurses that they feel frustration and guilt about not being able to fulfil expected gender roles (Theme 2), such as caring for family members and performing household chores. In accordance with the ‘gender perspective’ [45,46], the nurses described their patients as women who feel remorse and frustration at not being able to fulfil the gender expectation of caring in their culture, which involves exclusive responsibility for performing all housework tasks perfectly (including caring for other family members) [47]. Other authors have previously reported feelings of unease, an overload of work and family responsibilities, and domestic violence. In fact, previous studies have highlighted the importance of social support from other people with FM and from family members [48,49].

The participating nurses reported that the support of the family was very necessary for their FM patients. Patients frequently think that they do not receive this support from either the family or the healthcare system [48,49] (Theme 3). Studies have argued that patients with greater support from their social network have lower levels of anxiety and depression, which also helps to alleviate pain [50,51]. Support from one’s partner, therefore, can have a moderating effect on physical discomfort and help to control pain [46]. Shuster, McCormack, Pillai-Riddell, and Toplak [52] indicated that the lack of perceived family support of women with FMS may significantly impact their health outcomes.

We also found that problem-solving, decision-making, and communication skills were reported as important for managing chronic pain. Moreover, behavior patterns that encourage sufferers to face up to their condition help to minimize acute episodes of pain [53].

The nurses focused on problems more than on positive effects. This could be because in the GPST the first action performed was the identification of problems, and the solutions came later. In any case, this could also be because the nurses were convinced that they had suffered situations that had caused them great suffering, and that the patients had low self-esteem, and, therefore, it was difficult for them to solve problems. Additionally, the nurses believed that FM patients put others first, not themselves, who had carried a great physical and emotional burden on their backs and who needed empowerment. Some of them, but not the majority, had been able to successfully manage these aspects since they had managed to accept the disease and had the support of others. Many of them belonged to patient associations. In addition, over time, they had learned to seek resources to manage their illness and had managed to cope with them. Many of them had been able to overcome difficulties by first identifying the problems and seeing that the disease could be the consequence of learned behavior patterns and that they had to unlearn them to improve their emotions. At the same time, they realized how important communication with themselves and with the people around them was to express their desires and that others could offer their support.

### Strengths and Limitations

A possible limitation is that the sample was not large (i.e., the number of nurses running focus groups), but we do not believe that this is a limitation in qualitative research such as this, which seeks to understand a phenomenon rather than measure it. Consequently, the data analysis we present is informed by in-depth relational understanding developed from focus groups with highly experienced nurses working closely with FM patients. The analysis is further strengthened because it was conducted by a team of experienced healthcare professionals. Another possible limitation could be the use of the word trigger in the first question as it might not have been formulated in the most neutral way. However, our intention was that most of the elements that could be identified as stressors would be detected.

The results of this study carried out in a basically rural community in southern Catalonia, Spain, cannot be applied to patients from other cultures and/or countries. However, patients from similar cultures and environments may have more similarities than differences. It would be interesting to continue with cross-cultural research to examine differences and similarities between nurses’ perceptions of their FM patients.

This study mainly reflects the perceptions of female patients with FM reported by female nurses. This reflects the reality of FM and the nursing profession, both of which are predominantly female. However, in the future it would be interesting to carry out further research with a sample containing more men with FM and more male nurses. Overall, we suggest that this study is a worthwhile appreciation of the experience of FM today. What are the main problems you identify in people who have suffered from fibromyalgia? They can be both problems that triggered the disease or arising from any sphere of life and/or disease.

## 5. Conclusions

Throughout the study we have provided a detailed picture of the patients’ illness experiences from the point of view of professionals who determined the relationship between important factors in their lives and highlighted the key themes in the women’s experience of their illness as well as their social, employment, and family situations. The nurses observed that many of the women had suffered abuse not only in childhood and in the form of work overload but also in marriage since they had suffered infidelities and physical, psychological, and even economic abuse. They were not only abused by their partner but also by other relatives, who took advantage of their work. At first, they were not able to identify the importance of stress in the development of the disease, but little by little they realized the importance of emotions in their discomfort. Despite all this, it was difficult for them to express what they felt. They were also subject to considerable pressure from the gender roles in the society in which they had been brought up, so it was difficult for them to face up to their problems since they always sought the approval of others. Nurses found that their patients had considerable difficulty in managing stress and emotions and in identifying, expressing, and coping with everyday problems.

From our point of view, primary care nurses and FM unit nurses are frontline clinicians who have first-hand knowledge of the emotional problems of women who live with this poorly understood health condition on a day-to-day basis. As part of standard treatment, rehabilitation clinicians usually have the important task of providing comfort to patients in emotional distress without having been trained in specific psychological interventions. The illness can be better managed, and patients can adapt better if these strategies incorporate a gender approach since FM patients have a strong sense of obligation to fulfill what they consider to be their gender roles. Our findings about the perception of the disease will make it possible in the future to (1) develop interventions and programs that will enable clinicians to support their patients in their management of stressors and emotions; (2) enhance communication strategies; (3) identify and counter any pressure imposed by the cultural fulfilment of gender roles; and (4) improve family support. These programs should also empower patients and families, provide them with support regarding gender issues, and make them aware of when/whether they are being abused. We suggest that rehabilitation clinicians need to lead focused care for the empowerment of people and their families.

## Figures and Tables

**Table 1 healthcare-11-01531-t001:** Characteristics of the nurses that participated in the group discussions (n = 8).

Nurses	Age	Years in the Profession	Years in the Clinical Area	Workplace	Academic Level
Nurse 01	47	25	21	University	PhD
Nurse 02	57	35	8	University	MSc
Nurse 03	56	34	14	University health center	MSc
Nurse 04	40	18	18	University Mental health centre	MSc
Nurse 05	54	32	12	University	MSc
Nurse 06	55	33	13	University	PhD
Nurse 07	55	33	10	University	PhD
Nurse 08	21	0	0	University	Student

**Table 2 healthcare-11-01531-t002:** Characteristics of the FM patients participating in group-based problem-solving therapy.

Socio-Demographic Data	
**Gender (n females, %)**	42 (97.7%)
**Age, M (SD)**	61.1 (8.2)
**Employment status, nº (%)**	
Unemployed	3 (7.0%)
Active worker	3 (7.0%)
Paid employment but on sick leave	1 (2.3%)
Retired/pensioner	20 (46.5%)
Permanent disability	6 (14.0%)
Others	7 (16.3%)
Missing	3 (7.0%)
**Clinical Data (M, SD)**	
Time between the onset of symptoms and diagnosis (years)	9.8 (8.3)
Time living with the diagnosis (years)	11.6 (7.4)
Satisfaction with the healthcare system (0–10)	7.1 (3.1)
Satisfaction with medication (0–10)	5.2 (2.6)
Satisfaction with non-pharmacological treatments (0–10)	7.2 (2.9)

Note: One participant presented missing data in all study variables.

**Table 3 healthcare-11-01531-t003:** Themes and categories.

Themes	Categories
1. How stress and negative emotions can affect the body	It was common to find the presence of a “specific trigger” (stressful event) for FM symptoms.
	Some FM patients did not report a specific trigger but reported that they had endured harsh conditions throughout their lives.
	Patients consciously/unconsciously highlighted the importance of stress in FM.
	Patients were afraid to express what they feel.
	Gradually, the patients began to discuss the role played by the mind and how their emotional state influenced their pain.
2. Fulfilling expected gender roles	All patients put a very high limit on their daily activities because they have to assume their assigned gender role.
	The patients had difficulty in facing problems.
3. Lack of support from the family	Some of them received some support.
	In general, the patients received little support.
4. Abuse	The patients experienced physical, psychological, and economical abuse.
	Some patients found ways to manage and protect themselves from abuse.

## Data Availability

The datasets used and analyzed during this study are available from the corresponding author upon reasonable request.

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
