# Peer review of "Nurses’ Perceptions of Patient Fibromyalgia Illness Experiences after Performing Group-Based Problem-Solving Therapy: A Qualitative Research Study"

_healthcare, 2023, doi:10.3390/healthcare11111531_

Round 1
Reviewer 1 Report
Dear authors,
Congratulations on the study, the topics covered are interesting.
Minor considerations
Abstract
Line 18 - There is twice the information about the participants. both the results and the conclusion need to be rewritten.
Introduction
80–96% of all FMS patients1. - correct the reference style. ”The best way to treat FM symptoms is to adopt a multimodal approach, with pharmacological and non-pharmacological 36 strategies tailored to each patient [4]. In her recent study, Menzies [5] describes the latest 37 treatment regimens, pharmacological and non-pharmacological approaches, including alternative medicine, and nursing approaches designed to improve patient self-management.” - It should be mentioned which treatments are used, there are several recommendations for the use of exercise and psychological treatment with Cognitive Behavioral Therapy. Later you talk about Cognitive Behavioral Therapy, but nothing about exercise. Albuquerque, M. L. L., Monteiro, D., Marinho, D. A., Vilarino, G. T., Andrade, A., & Neiva, H. P. (2022). Effects of different protocols of physical exercise on fibromyalgia syndrome treatment: systematic review and meta-analysis of randomized controlled trials. Rheumatology International, 42 (11), 1893-1908. Vilarino, G. T., Branco, J. H. L., de Souza, L. C., & Andrade, A. (2022). Effects of resistance training on the physical symptoms and functional capacity of patients with fibromyalgia: a systematic review and meta-analysis of randomized clinical trials. Irish Journal of Medical Science (1971-), 1-14. Macfarlane GJ, Kronisch C, Dean LE, et al. (2017) EULAR revised recommendations for the management of fibromyalgia. Annals of the Rheumatic Diseases 76(2): 318–328. DOI: 10.1136/annrheumdis-2016-209724.
Materials and Methods
Line 89 - The participants in this study were nurses (n= 8) who were delivering GPST to FM patients. - Previously the authors have said that there are eight nurses. You should modify the text and avoid repeating information.
Lines 93-95 - “As shown in Table 1, the socio-demographic and clinical profile of the typical FM patient was a woman, approximately years of age, retired at the time of enrolment, diagnosed with FM for approximately 10 years, moderately to highly satisfied…” There is no table with patient information, only the participating nurses.
In my view, there is a big problem in the research structure:
Checking the perception of nurses about the experience of patients with fibromyalgia about their disease does not present great contributions either for patients or for nurses.
If the issue is the patients' experience, the best thing would be to do a survey asking them. If the question is the nurses' perception, the article should develop demonstrating how different perceptions can change the way nurses deal with patients. Which could have important practical implications.
Author Response
Response to Reviewer 1 Comments
Abstract:
Point 1: Abstract
Line 18 - There is twice the information about the participants. both the results and the conclusion need to be rewritten.
Thank you. Done
Point 2: Introduction
80–96% of all FMS patients1. - correct the reference style. Response: Done
”The best way to treat FM symptoms is to adopt a multimodal approach, with pharmacological and non-pharmacological 36 strategies tailored to each patient [4]. In her recent study, Menzies [5] describes the latest 37 treatment regimens, pharmacological and non-pharmacological approaches, including alternative medicine, and nursing approaches designed to improve patient self-management.” - It should be mentioned which treatments are used, there are several recommendations for the use of exercise and psychological treatment with Cognitive Behavioral Therapy. Done. Lines 59-61.
Later you talk about Cognitive Behavioral Therapy, but nothing about exercise. Albuquerque, M. L. L., Monteiro, D., Marinho, D. A., Vilarino, G. T., Andrade, A., & Neiva, H. P. (2022).
Effects of different protocols of physical exercise on fibromyalgia syndrome treatment: systematic review and meta-analysis of randomized controlled trials. Rheumatology International, 42 (11), 1893-1908.
Vilarino, G. T., Branco, J. H. L., de Souza, L. C., & Andrade, A. (2022). Effects of resistance training on the physical symptoms and functional capacity of patients with fibromyalgia: a systematic review and meta-analysis of randomized clinical trials. Irish Journal of Medical Science (1971-), 1-14.
Macfarlane GJ, Kronisch C, Dean LE, et al. (2017) EULAR revised recommendations for the management of fibromyalgia. Annals of the Rheumatic Diseases 76(2): 318–328. DOI: 10.1136/annrheumdis-2016-209724.
Added this references lines 69-73.
Point 3:
Materials and Methods
Line 89 - The participants in this study were nurses (n= 8) who were delivering GPST to FM patients. - Previously the authors have said that there are eight nurses. You should modify the text and avoid repeating information. We deleted repeating information.
Lines 93-95 - “As shown in Table 1, the socio-demographic and clinical profile of the typical FM patient was a woman, approximately years of age, retired at the time of enrolment, diagnosed with FM for approximately 10 years, moderately to highly satisfied…” There is no table with patient information, only the participating nurses. Sorry, we added the table 2.
In my view, there is a big problem in the research structure:
Checking the perception of nurses about the experience of patients with fibromyalgia about their disease does not present great contributions either for patients or for nurses.
If the issue is the patients' experience, the best thing would be to do a survey asking them. If the question is the nurses' perception, the article should develop demonstrating how different perceptions can change the way nurses deal with patients. Which could have important practical implications. Thank you, you are right, we did in previous article in men [22].
Our aim was to test nurses' perceptions so that they could be incorporated into clinical practice by improving routine clinical practice (the mind-body connections and the impact of stress on their symptoms, the expected gender roles, abuse and absence of social-family support ). For better understanding we add lines 112-118. An in the Abstract: line 17.

Reviewer 2 Report
Review “Nurses perception of FM illness experience”
This is an interesting paper describing a qualitative research study of the perception of illness experience of fibromyalgia (FM) patients by 7 nurses and a nursing student. The nurses had previously conducted group-based problem solving therapy (GPST) with FM patients and drew on their experience to describe themes.
The methodology is correct, the paper is well written and information provided will help to guide management of patients with FM.
Comments
1. The majority of references are 10 years old or older…Authors please review and provide more up to date references where available.
2. Line 43: uncertainty regarding diagnosis…this is too harsh…there is an international move to diagnose FM positively….please refer to AAPT criteria for diagnosis of FM as well as recent publications of Chronic primary pain conditions by IASP, as well as notion of nociplastic pain. Please revise.
3. Line 93: from where is this information derived…also not included in table 1.
4. Demographics of FM patients in GPST groups should be provided. The demographics would greatly influence the responses…eg age, relationships, working status.
5. Question 1 is a “leading” rather than an open ended question…specifically alluding to a “trigger” for FM, which is really only described for about 30% of patients. The authors must acknowledge this “leading question” in the discussion.
6. Although these focus groups address problem solving, I am disappointed that not a single positive effect was documented. Surely there must have been patients who were able to share coping or other strategies….is this context, it seems that the group sessions were merely a forum to complain. Please make comment to this effect in the discussion.
7. Please acknowledge in the discussion that these results may not be applicable to FM persons in other cultures/countries.
8. Supplementary materials were not accessible.
9. Ref 18 ..needs editing…it incorporates Ref 20

Author Response
Response to Reviewer 2 Comments
This is an interesting paper describing a qualitative research study of the perception of illness experience of fibromyalgia (FM) patients by 7 nurses and a nursing student. The nurses had previously conducted group-based problem-solving therapy (GPST) with FM patients and drew on their experience to describe themes.
The methodology is correct, the paper is well written, and information provided will help to guide management of patients with FM.
Point 1. The majority of references are 10 years old or older…Authors please review and provide more up to date references where available.
Response 1: Thank you. We have added three more references. Lines 34-40, 49-61, 368-389.
Point 2. Line 43: uncertainty regarding diagnosis…this is too harsh…there is an international sible limitación move to diagnose FM positively….please refer to AAPT criteria for diagnosis of FM as well as recent publications of Chronic primary pain conditions by IASP, as well as notion of nociplastic pain. Please revise.
Response 1: Done. Lines 34-40, 65-66.
Point 3. Line 93: from where is this information derived…also not included in table 1. Include
Response 3: Thank you, we add table 2.
Point 4. Demographics of FM patients in GPST groups should be provided. The demographics would greatly influence the responses…eg age, relationships, working status.
Response 4: Sorry, is not table1, is table 2. We have added in page 4.
Point 5. Question 1 is a “leading” rather than an open-ended question…specifically alluding to a “trigger” for FM, which is really only described for about 30% of patients. The authors must acknowledge this “leading question” in the discussion.
Response 5: We in the Discussion: Lines 368-389.
- Reumatologia
. 2022;60(4):242-246. doi: 10.5114/reum.2022.118677. Epub 2022 Sep 7.
Characterizing fibromyalgia flares: a prospective observational study
- Health Psychol Open
. 2018 Sep 25;5(2):2055102918802683. doi: 10.1177/2055102918802683. eCollection 2018 Jul-Dec.442-444.
What causes fibromyalgia? An online survey of patient perspectives.
Activation of acid-sensing ion channel 3 by
lysophosphatidylcholine 16:0 mediates psychological.
stress-induced fibromyalgia-like pain
Hyperbaric Oxygen Therapy Can Induce Neuroplasticity and Significant Clinical Improvement in Patients Suffering From Fibromyalgia With a History of Childhood Sexual Abuse-Randomized Controlled Trial
- doi: 10.3389/fpsyg.2018.02495. eCollection 2018.
Point 6. Although these focus groups address problem solving, I am disappointed that not a single positive effect was documented. Surely there must have been patients who were able to share coping or other strategies….is this context, it seems that the group sessions were merely a forum to complain. Please make comment to this effect in the discussion.
Response 6: We add lines 412-426.
Point 7. Please acknowledge in the discussion that these results may not be applicable to FM persons in other cultures/countries. Lines: 439-443.
Response 7: Done.
Point 8. Supplementary materials were not accessible.
Response 8: There are not supplementary files. Line 506-7.
Point 9. Ref 18 .needs editing…it incorporates Ref 20.
Response 9: Done

Round 2
Reviewer 1 Report
Dear authors,
Congratulations on the research.
The suggested changes have been made. I have no further comments.